# Pretreatment Affects Activated Carbon from Piassava

**DOI:** 10.3390/polym12071483

**Published:** 2020-07-02

**Authors:** Jonnys Paz Castro, João Rodrigo C. Nobre, Alfredo Napoli, Paulo Fernando Trugilho, Gustavo H. D. Tonoli, Delilah F. Wood, Maria Lucia Bianchi

**Affiliations:** 1Department of Forest Science (DCF), Federal University of Lavras, C.P. 3037, Lavras 37200-000, Brazil; trugilho@dcf.ufla.br (P.F.T.); gustavotonoli@yahoo.com.br (G.H.D.T.); 2Department of Forest Products (DPF), Forest Institute (IF), Federal Rural University of Rio de Janeiro, Rodovia BR 465, Km 07, C.P. 74527, 23890-000 Seropédica, Brazil; 3Center for Natural Sciences and Technology (CCNT), State University of Pará, Rodovia PA-125, s/n, Paragominas 68625-000, Brazil; rodrigonobre@uepa.br; 4Biomass, Wood, Energy, Bioproducts Research Unit, CIRAD, 73 Rue Jean François Breton, 34398 CEDEX5 Montpellier, France; alfredo.napoli@cirad.fr; 5Bioproducts Research, USDA ARS WRRC, Albany, CA 94710, USA; 6Department of Chemistry (DQI), Federal University of Lavras, C.P. 3037, Lavras 37200-000, Brazil

**Keywords:** piassava, Bahia, Amazon, *Attalea funifera*, *Leopoldinia piassaba*, corona discharge, electrical discharge, agricultural residues

## Abstract

The specificity of activated carbon (AC) can be targeted by pretreatment of the precursors and/or activation conditions. Piassava (*Leopoldinia piassaba* and *Attalea funifera* Martius) are fibrous palms used to make brushes, and other products. Consolidated harvest and production residues provide economic feasibility for producing AC, a value-added product from forest and industrial residues. Corona electrical discharge and extraction pretreatments prior to AC activation were investigated to determine benefits from residue pretreatment. The resulting AC samples were characterized using elemental analyses and FTIR and tested for efficacy using methylene blue and phenol. All resulting AC had good adsorbent properties. Extraction as a pretreatment improved functionality in AC properties over Corona electrical discharge pretreatment. Due to higher lignin content, AC from *L. piassaba* had better properties than that from *A. funifera*.

## 1. Introduction

Activated carbons (ACs) are widely used in many environmental remediation processes because of their high adsorption capacity. ACs can remove a wide variety of pollutants in aqueous environments by having large surface presence of functional groups with affinities for various adsorbates [1]. The use of ACs in effluent purification improved the taste, smell, color, UV absorbance and oxidability of treated water [2]. Various raw materials, including wood, bone, coconut shells, coconut endocarp, sugarcane bagasse and fruit seeds [3,4] have been used to produce ACs with different characteristics. The choice of precursors and activation conditions make it possible to design ACs for specific applications [5]. All carbonaceous feedstock has the potential to be used for AC manufacture but not all are economically feasible mostly because of the expense of gathering and transporting the feedstock if left in the field. If the feedstock is already consolidated into a relatively small area, such as a lumber mill or processing plant, the feedstock may be readily gathered, transported and treated. Better still would be to have localized plants set up to do the treatment.

Piassava fibers have great economic importance in Brazil, mainly in the states of Bahia and Amazonas, where large-scale production generates large quantities of residues. The residues are typically discarded or burned in boilers for energy generation [6].

AC production from solid residues using thermal conversion could divert a portion of the otherwise problematic byproduct away from the waste stream [7]. The AC industry has used, in the last decades, some agricultural and industrial residues as precursors with the objective of valorizing these raw materials or coproducts [5].

Piassava fiber has a smooth and impermeable texture due to its chemical composition, which can influence the properties of the ACs. Thus, the study of the possible effects of fiber pretreatment may indicate the best way to produce ACs with specific physicochemical characteristics. Avelar et al. [1], who worked on Bahia piassava fibers (*Attalea funifera* Martius) without pretreatment, stated that piassava fibers are a good precursor for AC production. Castro et al. [6] surveyed the effect of pretreatments including mercerization, corona discharge and extraction on long, unmilled Amazon piassava fibers and verified that pretreatments affect the properties of the resulting ACs. The aim of this work was to validate the novel pretreatments by including a second species of piassava and limiting the study to two pretreatments including Corona electrical discharge and solvent extraction on milled and screened piassava residues (*Leopoldinia piassaba* and *A. funifera*) on the resulting AC properties to provide validation of the previous work [6] and to do a more comprehensive evaluation of the resulting ACs.

## 2. Materials and Methods

### 2.1. Materials

Members of two genera of palm fiber, Amazon piassava (*L. piassaba*) (AP) and Bahia piassava (*A. funifera*) (BP), were used in this study. AP fibers were residual biomass from forest harvests in São Gabriel da Cachoeira (Brazil). BP fiber residues were donated by the broom industry in João Monlevade (Brazil). AP and BP fibers were milled and screened to 60-mesh for most purposes and 270 mesh for elemental and infrared analyses as per requirements of the experimental protocol.

### 2.2. Pretreatment of the Piassava Fibers

Prior to carbonizing the samples, milled piassava fibers were untreated (AP-Un, BP-Un), subjected to Corona electrical discharge (AP-Co, BP-Co) or solvent extraction (AP-Ex, BP-Ex) pretreatments. 

### 2.3. Electrical Discharge Pretreatment

AP and BP milled fibers were subjected to the electrical discharge produced by a Model PT-1 Corona Plasma Tech instrument (Corona, Brazil). A voltage of 10 kV was applied to the fibers for 10 min at a distance of 2 cm between the sample and the discharge head. Corona pretreated samples were termed AP-Co and BP-Co.

### 2.4. Extraction Pretreatment

Milled AP and BP fibers were wrapped in filter paper (seed germination filter paper, 60 g m^−2^) to prevent loss of material and treated with a 2:1 toluene:ethanol solution for 8 h using Soxhlet extraction. The toluene:ethanol solution was removed and replaced with ethanol and extracted for an additional 6 h. The samples were washed continuously with hot distilled water for 3 h and oven dried at 103 ± 2 °C for 24 h [8] resulting in extracted AP-Ex and BP-Ex.

### 2.5. Chemical Analysis of Untreated Precursor Material

Chemical analyses of the fibers of AP-Un and PB-Un were determined according to the standard methods: Holocellulose [9], cellulose [10], insoluble lignin [11], acid-soluble lignin [12] and ash [13]. Hemicellulose content was determined by the difference between holocellulose and cellulose contents.

### 2.6. Activated Carbon (AC) Preparation

ACs were produced from using 60-mesh untreated and pretreated piassava fibers. All materials were treated in a Fornitec F3-DM/T muffle furnace (Labnano, Rio de Janeiro, Brazil) at a heating rate of 100 °C h^−1^ to 550 °C and maintained for 1 h, resulting in charcoal (AP-Ch, BP-CH). AP-Ch and BP-Ch were moved to a cylindrical chamber in a tubular furnace (Sanchis Industrial Furnaces, Model 2023, Porto Alegre, Brazil), and activated at 800 °C (heating rate of 10 °C min^−1^) for 2 h in carbon dioxide environment at a flow rate of 150 mL min^−1^ resulting in 12 AC-treated samples: AP-Un-AC, BP-Un-AC, AP-Co-AC, BP-Co-AC, AP-Ex-AC and BP-Ex-AC.

### 2.7. Elemental Analysis

Elements (CHNS) of AP and BP samples from all steps were determined in an elemental analyzer (Vario MacroCube, Elementar Americas, Inc., Ronkonkoma, NY, USA), following the protocol described by Paula et al. [14]. Samples milled to 270 mesh, as specified in the protocol, were analyzed. Oxygen content was calculated by difference according to Equation (1).
O (%) = 100 − C (%) − H (%) − N (%) − S (%) − ash (%)(1)
where O is oxygen, C is carbon, H is hydrogen, N is nitrogen and S is sulfur [13].

### 2.8. Infrared Spectroscopy (FTIR)

Specific functional groups of the 12 samples of Ch and ACs made from 270 mesh AP and BP were identified using Fourier transform infrared (FTIR) spectroscopy analysis (Digilab Excalibur FTS 3000, Bio-Rad, Hercules, CA, USA) in the spectral range of 400 to 4000 cm^−1^ and 4 cm^−1^ resolution. 

### 2.9. Adsorption Tests and Modeling 

The adsorption isotherms of methylene blue dye and phenol were obtained using 10 mg of AC adsorbent and 10 mL of adsorbate at varying concentrations (25, 50, 100, 250, 500 and 1000 mg L^−1^). AC-adsorbate mixtures were stirred at 100 rpm for 24 h at room temperature (25 ± 2 °C). The equilibrium concentration was determined in a UV–visible spectrophotometer (AJX-3000PC, AJ Lab, AJ Micronal, Sao Paulo, Brazil) at of 665 nm and 270 nm for methylene blue and phenol, respectively. The Langmuir (Equation (2)) and Freundlich (Equation (3)) isotherm models [15] were used to analyze the data:q_eq_ = (q_m_ K_L_ C_eq_)/(1 + K_L_ C_eq_)(2)
where q_eq_ is the equilibrium concentration of AC (mg g^−1^), C_e_ is the equilibrium concentration in the solution (mg L^−1^), q_m_ is the maximum capacity of adsorption of the AC (mg g^−1^) and K_L_ is the Langmuir adsorption constant (L mg^−1^).
q_eq_ = K_F_ C_e_^(1⁄n)^(3)
where K_F_ is a constant that indicated the relative capacity of adsorption (mg g^−1^) (L g^−1^)^−(1/n)^ and n is related to the intensity of adsorption of the AC.

### 2.10. Iodine Number (IN) and Surface Area with Methylene Blue–S_MB_

The iodine number, defined as the absorption of iodine in mg g^−1^ AC, was determined according to ASTM D4607-14 [16].

The surface area estimated with methylene blue (S_MB_) was obtained as reported in Equation (4) [17]:S_MB_ = S_MB_° q_m_(4)
where S_MB_° is the methylene blue surface area (1.93 m^2^ mg^−1^) and q_m_ is the maximum capacity of adsorption of the AC (mg g^−1^).

### 2.11. Estimation of the Brunauer, Emmett and Teller (BET) Surface Area

The micropore volume, total pore volume and AC surface areas were estimated from the IN and maximum adsorption of methylene blue (q_m_). Structural Characterization of Activated Carbon (SCAC) software was used with the data to estimate the micropore volume, the total pore volume and the BET surface area of ACs [18].

## 3. Results and Discussion

### 3.1. Chemical Composition of Fibers

The chemical compositions of the untreated AP and BP used in AC preparation are described in Table 1. AP and BP have low ash and high lignin contents which are important characteristics in producing AC with optimal properties. High lignin contents provide high yields thereby contributing to the fixed carbon content at the end of carbonization and activation processes. Lignin biochar was shown to have a much higher yield than cellulose or pine wood [19] probably due to the higher decomposition temperature of lignin. Decomposition of hemicellulose, cellulose and lignin occur in the range of 200–260, 240–350 and 280–500 °C, respectively [20]. Lignin decomposes at higher temperatures and over a much wider temperature range than do hemicellulose or cellulose; the aromatic rings of lignin make it much less susceptible to degradation and require high temperatures.

AC derived from AP had higher lignin and extractives and lower cellulose, hemicelluloses and ash than AC derived from BP. The results found in this work, for the chemical composition of BP-Un, agree with those obtained by Schuchardt et al. [21], who found values of 0.8, 0.7, 45.0, 28.6 and 25.8 for ash, extractives, lignin (Klason), cellulose and hemicellulose, respectively.

### 3.2. Elemental Composition

Elemental composition of the piassava fibers, charcoal and activated carbon are shown in Table 2. AP had higher lignin (Table 1) and N (Table 2) contents that BP. However, after the pretreatments by Corona discharge or extraction, the two samples had similar values.

AP-Un-Ch had higher C than BP-Un-Ch. The comparative increase in C and the resulting decrease in H and O during thermal conversion are shown in Table 2. High heat treatment volatilizes H-, O- and N-containing compounds [22] and results in high C contents. Pretreatments mostly affected the elemental composition of Ch (AP-Un-Ch, AP-Co-Ch, AP-Ex-Ch, BP-Un-Ch, BP-Co-Ch and BP-Ex-Ch).

In addition to the elements reported here, silica is also found in both AP and BP and was found to be 52% in *A. funifera* in the form of SiO_2_ [23] and occurred as bodies with protrusions of about 20 µm in diameter taking up most of the space in a single cell [24,25]. Energy dispersive X-ray analysis showed that the bodies were mostly of silica [26].

### 3.3. Infrared Spectroscopy (FTIR)

FTIR was used to identify specific functional groups in AP-Un and BP-Un and compare the results with the pretreated samples and with the charcoal and activated carbons obtained from the piassava fibers (Figure 1).

The FTIR spectra between 2000 and 500 cm^−1^ best describes functional groups in the samples. However, countless peaks in this region due the stretching and deformational vibrations of functional groups belonging to lignin and carbohydrates create overlapping peaks that make interpretation difficult. Close inspection reveals that AP-Un and BP-Un spectra (Figure 1a) have peaks at 1715 cm^−1^ characterizing C=O (unconjugated ketone, conjugated carboxylic acids groups and ester groups of carbohydrates), 1600 cm^−1^ representative of C=C (aromatic ring of lignin) plus C=O, 1510 cm^−1^ attributed to lignin backbone, 1270 cm^−1^ as lignin guaiacyl ring plus C=O and 1055 cm^−1^ assigned to C–O(H) of carbohydrates [27,28,29].

In BP-Un and AP-Un (Figure 1a) the peaks at 1270 and 1055 cm^−1^ have about the same intensity, however, the lignin peaks, notably the peak at 1510 cm^−1^, are more intense in AP-Un (Figure 1a) than in BP-Un. The greater intensity of the lignin peaks confirms that AP-Un contains more lignin than BP-Un (Table 1).

According to Sadeghnejad et al. [30], corona treatment induces changes in material surface properties mostly due to reactions between covalent bonds in atmospheric oxygen and carbons on the sample surface. However, such change is not obvious in the spectra (Figure 1b) perhaps because corona treatment occurs on material surfaces, therefore, once the material has been ground for measurement by FTIR, the differences are no longer apparent in the resulting spectra.

A peak at 1560 cm^−1^, attributed to C=C of the aromatic skeleton may be seen in charcoal spectra in both AP-Un-Ch and BP-Un-Ch (Figure 1d). The 1560 cm^−1^ may be due to an increase in carbonization that results in dehydrogenation and the formation of double bonds that indicated aromatization of the material [27,28]. Table 2 presents data that confirm the decrease in hydrogen in the charcoals. 

The 1050 cm^−1^ peak disappears in the charcoal spectra (Figure 1d–f) indicating carbohydrate degradation. A peak appears at about 1190 cm^−1^ showing that the aromatic structure of lignin has changed due to carbonization. (Yang et al. 2007).

Neither pretreatment had a significant effect on charcoal or activated carbons (Figure 1d–i).

### 3.4. Adsorption Tests

Figure 2 and Figure 3 show the isotherms that can be classified as type I (Langmuir type). According to the International Union of Pure and Applied Chemistry (IUPAC) classification, the Langmuir type of isotherm indicates high affinity between adsorbate and adsorbent. The adsorption isotherm is an important tool that reveals the specific relationship between the adsorbate molecules and the carbonaceous adsorbents [31]. The Langmuir and Freundlich parameters, for methylene blue and phenol (Table 3) show that the piassava ACs adsorbed higher quantities of methylene blue than phenol and that there was an influence from the pretreatments in the adsorption processes of the two compounds. The Langmuir model assumes a monolayer coverage of dyes over a homogenous adsorbent surface without further adsorption occurring in the occupied sites. In comparison, a heterogeneous adsorption was described by the Freundlich isotherm. The Freundlich model showed a continued increase in the concentration of dye on the adsorbent surface [32] and did not assume homogenous site energies or limited adsorption.AP-AC have higher adsorption capacity for methylene blue than BP-AC. AC produced from extracted AP and BP (AP-Ex-AC and BP-Ex-AC) had higher adsorption capacity than other pretreatments tested. The Langmuir model showed a better fit to the isotherm data than the Freundlich model as indicated by the R^2^ (Qe vs. Ce) values. The better fit Langmuir model indicated that adsorption occurred on a homogeneous surface [33].

The R^2^ (Qe vs. Ce) values for adsorption of phenol by AP-AC fit well to the Langmuir model. BP-Un-AC fit well to the Freundlich model whereas BP-Co-AC and BP-Ex-AC fit well to the Langmuir model. AP-Un-AC and BP-Un-AC had the highest adsorption capacity values for phenol compared to other ACs. ACs produced with Corona pretreated fibers (AP-Co-AC and BP-Co-AC) had decreased adsorption compared to other ACs. The adsorption capacity is also influenced by surface chemistry probably due to the presence of superficial functional groups and heteroatoms in the ACs. Chemical composition may be an additional factor affecting the surface chemistry of the ACs. Several factors may contribute to the adsorption capacity of phenol in addition to the best fit model that is provided to explain its adsorption. The effective adsorbent capacity is determined by the initial concentration of the adsorbent, ionic strength, temperature and pH, all of which significantly affect phenol removal efficiency [34,35]. Xiong et al. [36] showed that phenol was removed more efficiently at an alkaline pH during the initial 25 min of reaction time and leveled off thereafter. Beker, et al. state that surface pH of adsorbents decreases with increasing degree of activation and that with increasing alkalinity, a reduction in adsorption capacity results. The optimal pH of 6.5 was found for adsorption of phenol. However, pH affects the type and ionic state of surface functional groups [37], so adsorbance capacity depends on the functional groups on the activated carbon. In this study, pH was not determined, however, FTIR results indicate a slightly acidic pH of AC due to the presence a strong peak at about 2339 cm^−1^ (Figure 1) indicative of stretching of CO_2_ [38].

Piassava AC in this study was shown to have greater capacity to adsorb methylene blue (223–424 mg g^−1^) compared to other ACs found in literature [1,6,32,39,40,41,42]. AC adsorption for phenol in this study (125–226 mg g^−1^) were similar to reported values in the literature (145–278 mg g^−1^) [1,6,43,44].

Avelar et al. [1] showed that activated carbons produced from piassava fibers have high adsorption capacity for specific compounds, most often better than commercial AC used for comparison in the study of these authors.

Castro et al. [6] reported a q_m_ of 388 for adsorption of methylene blue (Corona pretreatment), whereas, in the present work, a lower q_m_ value of 385 is reported. The pretreatments for the two studies were slightly different which is perhaps the reason for the numerical differences in adsorption of methylene blue. In Castro et al. [6], the fibers were Corona pretreated whole whereas, in this study, the fibers were ground prior to Corona pretreatment. Perhaps in the previous work, the application of Corona pretreatment on whole fibers was more homogeneous than grinding the fibers prior to pretreatment. 

### 3.5. Surface Area of the ACs

The iodine number, methylene blue adsorption and BET (Est-S_BET_) surface area estimations of AC are shown in Table 4. The iodine number and the methylene blue number are important and can be used to estimate of the surface area, of the micropore volume and of the total pore volume of activated carbon samples through multiple regression, thus, the two parameters are important since the materials possess different pore sizes which can be accessed by the different molecules according to the pore geometry [18]. IN is the capacity of the adsorbent (AC) to decolorize a compound and indicates adsorbent microporosity [45] and degree of activation [39]. Methylene blue adsorption studies are widely used to evaluate adsorbents. Due the dimensions of the methylene blue molecule, the dye serves as a visible marker and an indicator of mesoporosity and, to a lesser extent, microporosity. The iodine molecule is relatively small and is, therefore, an indicator of microporosity [18]. The development of pore structure in the AC during activation followed three main steps: (1) opening of unreachable pores, (2) creation of new pores and (3) enlargement of existing pores [31]. 

Pretreatment of AP and BP resulted in changes to the surface areas and pore volumes. In general, the AC produced from extracted pretreatment (AP-Ex-AC, BP-Ex-AC) had the largest surface areas of all ACs. During the extraction process, soluble phenolic and carbohydrate components were removed, eliminating pore obstruction by those components during the carbonization and activation processes. Corona electrical discharge pretreatment also showed potential to alter and improve the qualities of the resulting pretreated ACs.

The differences between AP-AC and BP-AC seems to be associated with the chemical composition of the untreated raw materials (Table 2) which influences the pore formation and surface areas of the resulting ACs. AP has higher lignin content than BP. Based on the findings of [46], one would expect a decrease in surface area with an increase in lignin content since their conclusion is that due to the large numbers of oxygen-containing functional groups in cellulose and hemicellulose, the small molecules (H_2_O, CO_2_ and CO) would volatilize during heat treatment and be replaced by micropores. They state that since lignin has aromatic units that are chemically inert, a nonporous AC would be produced. However, Deng et al. [46] used chemical activation and treated at 400 °C, at which temperature lignin would not decompose. In this study, activation was with CO_2_ at 800 °C, above the temperature that lignin would start to decompose. Delignification of Elmwood was used to release sugars of hemicellulose for production of value-added products, a high-lignin fraction resulted from the process and was converted into an additional high-value microporous AC of high surface area (1220 m^2^g^−1^) [47].

The estimated surface areas (Est-S_BET_) of ACs in the present study were lower than those reported in the literature that used chemical activation [1,31,32,33,39]. Chemical activation typically yields AC with higher surfaces area than physical activation. Values of the same magnitude were found when compared with studies that performed physical activation [1,6,45]. Maneerung et al. [45] reported SBET values of 737 m^2^ g^−1^ from woody biomass after steam or CO_2_ activation at 800 °C with corresponding yields of 55% and 39%, respectively. The starting material was residual charcoal from mesquite (*Prosopis* spp.) chips which had been processed for syngas. Thus, the feedstock precursor was very different in the Maneerung et al. [45] study than that in this study in that there was probably little or no volatile compounds remaining associated with the charcoal that could get redeposited on the carbon [6] effectively clogging or reducing the pore diameters or numbers. The conditions for activation were for 3 h at a gas flow rate of 90 mL min^−1^ in Maneerung et al. [45] vs. 2 h at a flow rate of 150 mL min^−1^ in this study. Activation at elevated temperatures for long duration, generally results in a reduction in AC yield. According to Maneerung et al. [45], AC surface area increases with increasing process temperature and with the amount of activating agent.

In AP-AC and BP-AC, the chemical differences between the piassavas and the two pretreatments (Corona electrical discharge and extraction) influenced the final AC properties. However, the AP-AC and BP-AC had properties similar to those found in the literature. The ACs of this work were physically activated, which is an advantage, since there is no need for extra washing or the use of expensive reagents.

## 4. Conclusions

ACs prepared from Amazon and Bahia piassava fibers, with and without pretreatment, by physical activation with CO_2_ have potential for use as adsorbent. The pretreatments applied to the fibers influenced the properties of the ACs.

The initial chemical compositions of Amazon and Bahia piassavas affect the physical-chemical characteristics of the resulting AC due to the higher lignin content in AP compared to BP. The ACs had higher adsorption capacity of methylene blue than phenol, according to adsorption isotherms and Langmuir and Freundlich parameters. Extraction as a pretreatment improved AC properties over Corona electrical discharge pretreatment. All of the ACs in this work had good adsorbent properties, employing extraction as a pretreatment was an improvement over values reported in the literature and AP-AC had better properties than BP-AC. 

This study provides relevant information on the characteristics of the ACs produced from untreated and pretreated piassava fibers and verifies previously reported research.

## Figures and Tables

**Figure 1 polymers-12-01483-f001:**
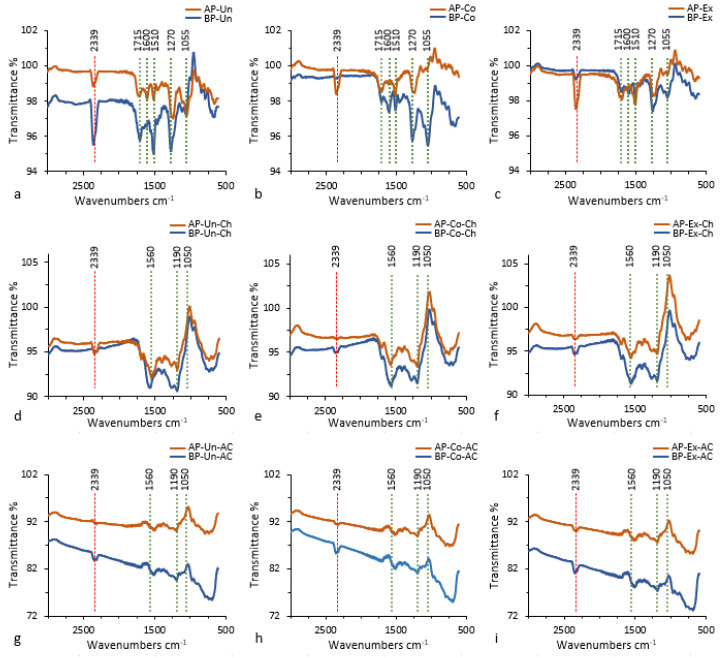
Typical infrared spectra of *Leopoldinia piassaba* and *Attalea funifera* Martius raw, milled fibers (**a**–**c**), charcoal precursor (**d**–**f**) and activated carbon (**g**–**i**). Fibers without pretreatment (**a**,**d**,**g**), fibers pretreated by Corona electrical discharge (**b**,**e**,**h**) and fibers pretreated by solvent extraction (**c**,**f**,**i**). The vertical lines (**a**–**c**) correspond to specific functional groups: 1715 cm^−1^ (C=O), 1600 cm^−1^ (C=C of the aromatic ring of lignin plus C=O), 1510 cm^−1^ (lignin backbone), 1270 cm^−1^ (lignin guaiacyl ring plus C=O) and 1055 cm^−1^ (C-O(H) of carbohydrates). Note that BP-Un and AP-Un (a) have the same intensity at peaks 1270 and 1055 cm^−1^. The peak at higher intensity at the peak 1510 cm^−1^ in AP-Un compared to BP-Un is indicative of higher lignin in AP-Un than in BP-Un. The peaks at 1560 cm^−1^, attributed to C=C of the aromatic skeleton, indicates dehydrogenation, the formation of double bonds and aromatization of the material in AP-Un-Ch and BP-Un-Ch). The 1050 cm^−1^ peak has disappeared (**d**,**e**), suggesting carbohydrate degradation. The peak at 1190 cm^−1^ shows the change of aromatic structure of the lignin due to the carbonization. The pretreatments had no significant effect on either charcoal or activated carbons (**d**–**i**). AP, *L. piassaba* (Amazon piassava); the peaks at 2339 correspond to CO_2_ stretching (**a**–**i**). BP, *A. funifera* (Bahia piassava); Ch, charcoal; AC, activated carbon; Un, untreated (no pretreatment); Co, Corona electrical discharge pretreatment; Ex, solvent extraction pretreatment.

**Figure 2 polymers-12-01483-f002:**
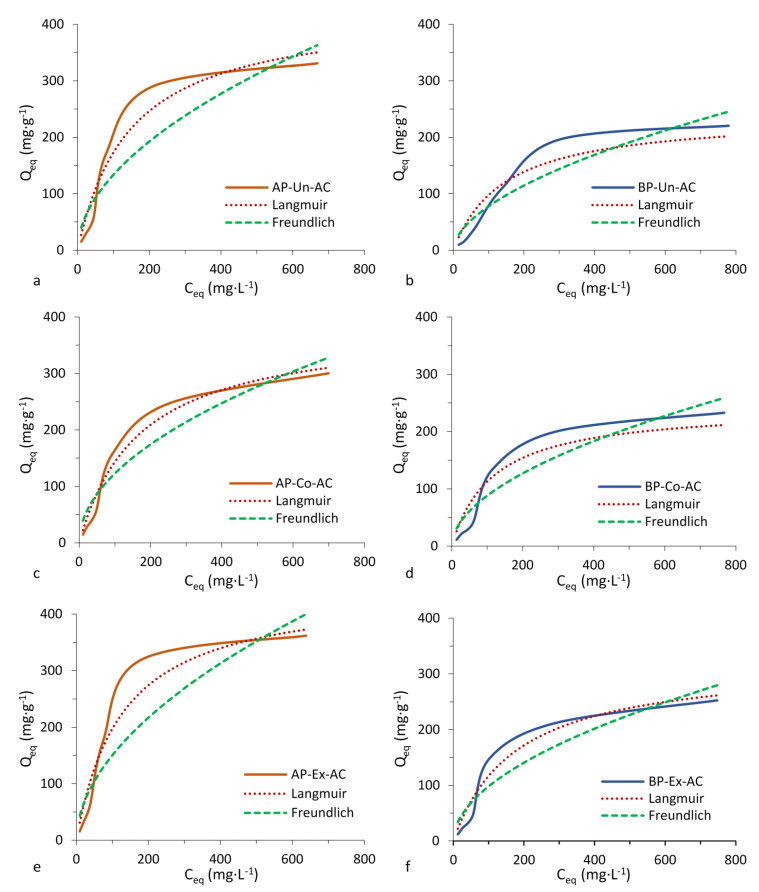
Methylene blue dye adsorption isotherms of piassava activated carbon with and without pretreatments comparing Langmuir and Freundlich curves. AP, Amazon piassava; BP, Bahia piassava; Ch, charcoal; AC, activated carbon; Un, untreated, no pretreatment; Co, Corona electrical discharge pretreatment; Ex, solvent extraction pretreatment.

**Figure 3 polymers-12-01483-f003:**
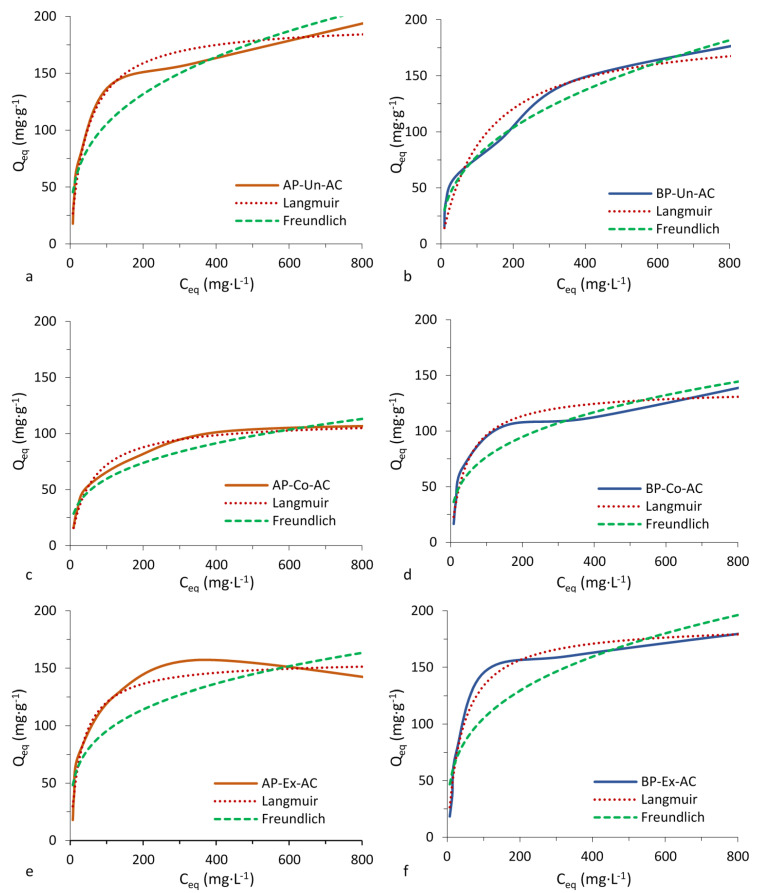
Phenol adsorption isotherms of piassava activated carbons with and without pretreatments comparing Langmuir and Freundlich curves. AP, Amazon piassava; BP, Bahia piassava; Ch, charcoal; AC, activated carbon; Un, untreated, no pretreatment; Co, Corona electrical discharge pretreatment; Ex, solvent extraction pretreatment.

**Table 1 polymers-12-01483-t001:** Chemical composition of the of *Leopoldinia piassaba* (AP) and *Attalea funifera* Martius (BP).

Fiber Source	Chemical Composition (%)				
	Cellulose	Hemicellulose	Lignin	Extract	Ash
AP	19.30 ± 0.08	20.83 *	55.86 ± 0.36	3.41 ± 0.23	0.59 ± 0.02
BP	27.51 ± 0.08	26.03 *	45.93 ± 1.49	1.48 ± 0.09	0.76 ± 0.08

* Calculated by difference.

**Table 2 polymers-12-01483-t002:** Elemental composition of piassava fibers (untreated and pretreated), charcoal and activated carbons.

Samples.	C (%)	O (%)	N (%)	H (%)
AP-Un	53.13	40.07	1.57	5.22
AP-Co	52.24	41.24	1.37	5.14
AP-Ex	52.77	40.66	1.34	5.22
BP-Un	50.61	42.85	1.15	5.39
BP-Co	52.28	41.18	1.05	5,49
BP-Ex	52.76	40.74	1.01	5.49
AP-Un-Ch	78.22	16.76	2.42	2.57
AP-Co-Ch	74.34	20.51	2.34	2.79
AP-Ex-Ch	73.37	21.24	2.79	2.57
BP-Un-Ch	69.44	26.69	1.38	2.47
BP-Co-Ch	79.50	16.27	1.47	2.75
BP-Ex-Ch	78.23	17.63	1.41	2.72
AP-Un-AC	82.03	13.53	2.52	1.82
AP-Co-AC	84.48	11.86	1.85	1.77
AP-Ex-AC	82.66	13.51	1.96	1.85
BP-Un-AC	83.38	13.66	1.39	1.52
BP-Co-AC	81.37	15.65	1.37	1.51
BP-Ex-AC	82.45	14.79	1.40	1.27

C, Carbon; O, Oxygen; H, Hydrogen; N, Nitrogen; AP, Amazon piassava (*Leopoldinia piassaba*); BP, Bahia piassava (*Attalea funifera* Martius). Pretreatments: Un, untreated; Co, Corona electrical discharge; Ex, extraction; Ch, charcoal; AC, activated carbon.

**Table 3 polymers-12-01483-t003:** Langmuir and Freundlich parameters for the adsorption of methylene blue and phenol by the activated carbons obtained in this study.

Compound	Activated Carbons (AC)	Langmuir Parameters			Freundlich Parameters		
		q_m_	K_L_	R^2^	K_F_	1/n	R^2^
Methylene blue	AP-Un-AC	427	0.007	0.96	12	0.526	0.87
	AP-Co-AC	385	0.006	0.98	12	0.504	0.92
	AP-Ex-AC	446	0.008	0.96	13	0.527	0.86
	BP-Un-AC	239	0.007	0.93	6	0.560	0.90
	BP-Co-AC	243	0.009	0.93	8	0.529	0.90
	BP-Ex-AC	324	0.006	0.96	9	0.521	0.90
Phenol	AP-Un-AC	194	0.022	0.99	24	0.319	0.93
	AP-Co-AC	112	0.018	0.99	14	0.307	0.95
	AP-Ex-AC	156	0.033	0.98	29	0.258	0.83
	BP-Un-AC	192	0.008	0.96	12	0.405	0.98
	BP-Co-AC	137	0.023	0.97	19	0.304	0.93
	BP-Ex-AC	188	0.025	0.99	26	0.300	0.88

q_m_ = maximum quantity of adsorption (mg g^−1^); KL = Langmuir constant (L mg^−1^); R^2^ = correlation coefficient; K_F_ = Freundlich constant (mg g^−1^) (L g^−1^)^−1/n^; 1/n = Freundlich parameter. AP, Amazon piassava; BP, Bahia piassava; Ch, charcoal; AC, activated carbon; Un, untreated, no pretreatment; Co, Corona electrical discharge pretreatment; Ex, solvent extraction pretreatment.

**Table 4 polymers-12-01483-t004:** Textural characterization parameters of the activated carbons from piassava with different pretreatments.

Activated Carbon	S_AM_ (mg^2^ g^−1^)	IN (mg g^−1^)	Est-S_BET_ (mg^2^ g^−1^)	V_total_ (cm^3^ g^−1^)	V_micro_ (cm^3^ g^−1^)
AP-Un-AC	824	508	611 ± 67	0.99 ± 0.13	0.98 ± 0.16
AP-Co-AC	743	514	597 ± 65	0.91 ± 0.12	0.79 ± 0.13
AP-Ex-AC	861	572	679 ± 74	0.98 ± 0.13	0.95 ± 0.15
BP-Un-AC	461	525	539 ± 59	0.64 ± 0.08	0.31 ± 0.05
BP-Co-AC	469	539	552 ± 60	0.65 ± 0.08	0.32 ± 0.05
BP-Ex-AC	625	575	619 ± 68	0.81 ± 0.10	0.57 ± 0.09

S_MB_ = surface area estimated of methylene blue; IN = iodine number; Est-S_BET_ = estimated BET surface area; V_total_ = estimated total pore volume; V_micro_ = estimated micropore volume. AP, Amazon piassava; BP, Bahia piassava; Ch, charcoal; AC, activated carbon; Un, untreated, no pretreatment; Co, Corona electrical discharge pretreatment; Ex, solvent extraction pretreatment.

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
