# Peer review of "Pretreatment Affects Activated Carbon from Piassava"

_polymers, 2020, doi:10.3390/polym12071483_

Round 1

Reviewer 1 Report

The paper by Castro et al. describes the effect of corona electric discharge and extraction on AC's produced from Piassava (Leopoldinia piassaba and Attalea funifera Martius. The AC's are characterized for their physical and chemical properties.
The paper is well-written, it is rich in data and referencing is quite good. Most statements are supported by the data and authors do not generally over interpret or oversell their work. All in all, the paper is a good piece of work which could be published in Polymers.

However, one concern I have is with the Novelty of the research. Is it the pretreatment methods or their application as adsorbents? AC's from Piassava have been investigated and most recently by same authors: https://doi.org/10.1080/15440478.2018.1442280. Authors will have to highlight novelty in this research.  If this is successfully solved, the work constitutes an advance and it is suitable for Polymers. Otherwise, it is not.

Author Response

Thank you for the thorough review of our manuscript. We have added information to the Introduction and Conclusion which indicates that we are considering this the second work on pretreatments of piassava to be a more thorough investigation by limiting the conditions of pretreatment and by including one additional species of piassava. We also milled the fibers instead of keeping them whole, which supplied more fiber surface area to the pretreatments. We also have included more measurements to more thoroughly test the theory that pretreatments are effective.

Reviewer 2 Report

This is a clear presentation of the effect of various pretreatment methods on the properties of activated carbons obtained from two varieties of piassava lignocellulosic precursors. The presentation of methods and results is clear and appropriate. The differences between the properties of various products (evaluated from standard solution adsorption methods and analyzed according to classical methods) are clearly summarized.

A few issues that should be addressed before possible publication:

1 - The title is confusing, but this is not the first instance when these two words, "affects" and "effects", are confused. If using "affects" (= a verb, meaning the action to modify or change something) a correct title would be "PRETREATMENT AFFECTS ACTIVATED CARBON FROM PIASSAVA". If using the word "effects" (= a noun, meaning the results of a change) the correct title should be "PRETREATMENT EFFECTS ON ACTIVATED CARBON FROM PIASSAVA". 

2 - A careful revision of the text will show several instances where the superscripts (mostly related to units) and subscripts (in chemical formulae) are not properly indicated.

3 - The correct units for Freundlich constant are (mg g-1) (L g-1) -(1/n) (see line 144 and notes to table 4).

Author Response

  1. The title is confusing, but this is not the first instance when these two words, "affects" and "effects", are confused. If using "affects" (= a verb, meaning the action to modify or change something) a correct title would be "PRETREATMENT AFFECTS ACTIVATED CARBON FROM PIASSAVA". If using the word "effects" (= a noun, meaning the results of a change) the correct title should be "PRETREATMENT EFFECTS ON ACTIVATED CARBON FROM PIASSAVA". 
  • Response: Thanks for noting that. We have removed the work “on” from the title to clarify the intended meaning.
  1. A careful revision of the text will show several instances where the superscripts (mostly related to units) and subscripts (in chemical formulae) are not properly indicated.
  • Response: We’ve found several instances where super/subscripts were missing and hope that we’ve found all of them.
  1. The correct units for Freundlich constant are (mg g-1) (L g-1) -(1/n) (see line 144 and notes to table 4).
  • Response: Thank you for the thorough review of our manuscript. Thanks for pointing out the error. We hope that we have corrected all instances of super/sub scripts.

Reviewer 3 Report

This manuscript presents study of corona electrical discharge and extraction pretreatments prior to activated carbon (AC) activation to determine advantages from residue pretreatment. Two genera of palm fibers, Amazon piassava (Leopoldinia piassaba) (AP) and Bahia piassava (Attalea funifera) (BP), were used in the study. The elemental analyses and the resulting activated carbon samples were characterized using elemental analyses and Fourier transform infrared (FTIR) spectroscopy and tested for efficacy using methylene blue and phenol. All resulting AC had good adsorbent properties. It was shown that the extraction as a pretreatment improved functionality in AC properties over Corona electrical discharge pretreatment. Due to higher lignin content, AC from Leopoldinia piassaba had better properties than that from Attalea funifera.

All the results obtained are important and interesting. Therefore, no doubt I would recommend a manuscript for publication in Polymers.

Author Response

  1. All the results obtained are important and interesting. Therefore, no doubt I would recommend a manuscript for publication in Polymers.
  • Response: Thank you for the thorough review of the manuscript. No changes were made from this review.

Reviewer 4 Report

I have finished a review of the paper polymers-832943 titled: “Pretreatment affects on activated carbon from piassava” and written by the authors: Jonnys Paz Castro , João Rodrigo C. Nobre , Alfredo Napoli , Paulo Fernando Trugilho , Gustavo H. D. Tonoli , Delilah F Wood * , Maria Lucia Bianchi

In presented paper, new simple easy to perform ways were used for pretreatment of Leopoldinia piassaba and Attalea funifera Martius in order to obtain activated carbon materials which potentially may be used in the area of environmental protection and removal methylene blue and phenols from contaminated water solutions.

In general, the paper is good written, the topic is very actual, the English is correct. From that point I recommend Editor to accept it for publication. However, in the paper are some parts which must be changed and rewritten, and because of that I suggest its Major revision.

My objections are:

In 3.3 Results of FTIR analysis are presented quite confusingly and it is very difficult to follow the explanations and what is shown in the Figures. For this reason, the authors must clearly indicate in each Figure at which wave numbers are which FTIR band, and then through the text very precisely, band by band, explain from what it comes from and if there are changes, how these changes can be explained. The way at which are results presented in this version of the paper is not acceptable for me.

In 3.4. Results of the MB and phenols adsorption are presented and fitted with two isotherm models: Langmuir and Freundlich. However, for both models were obtained very low R2 values (mainly <0.90) and for some samples R2 was about 0.80. That means that applied isotherms do not fit in satisfying way presented results. That additionally means that all followed discussion is not correct and is wrong. From that reason, authors must apply other models and try to find that one which fits results much better and then make a new discussion based on that model or models. Publishing paper with discussion presented in 3.4 in this version of manuscript is not acceptable for me.

Also, pH is very important parameter when it comes to adsorption, nevertheless, authors did not followed influence of this that parameter on the adsorption of the both pollutants (MB and phenols). It is required from the authors to insert in the papers results of that investigations and give adequate explanations.

On other parts of the paper I do not have significant and important objections.

Best regards

Author Response

  1. In 3.3 Results of FTIR analysis are presented quite confusingly and it is very difficult to follow the explanations and what is shown in the Figures. For this reason, the authors must clearly indicate in each Figure at which wave numbers are which FTIR band, and then through the text very precisely, band by band, explain from what it comes from and if there are changes, how these changes can be explained. The way at which are results presented in this version of the paper is not acceptable for me.
  • Thank you for thorough review of our paper. We have made changes to the text, graph and graph caption. We hope that the explanation covers what is needed. The graphs have been reorganized to make discussion more relevant.
  1. In 3.4. Results of the MB and phenols adsorption are presented and fitted with two isotherm models: Langmuir and Freundlich. However, for both models were obtained very low R2 values (mainly <0.90) and for some samples R2 was about 0.80. That means that applied isotherms do not fit in satisfying way presented results. That additionally means that all followed discussion is not correct and is wrong. From that reason, authors must apply other models and try to find that one which fits results much better and then make a new discussion based on that model or models. Publishing paper with discussion presented in 3.4 in this version of manuscript is not acceptable for me.
  • We discovered some mistakes in the calculations and recalculated our values and obtained higher correlation values. The graphs, text and tables have been changed to reflect the correction.
  1. Also, pH is very important parameter when it comes to adsorption, nevertheless, authors did not followed influence of this that parameter on the adsorption of the both pollutants (MB and phenols). It is required from the authors to insert in the papers results of that investigations and give adequate explanations.
  • We regret that pH was not measured as part of this study and it is too late to recuperate the samples and measure pH. The person (JPC) who conducted that portion of the experiment has left the facility in which he worked and the samples are no longer available. However, we cited information that leads us to believe that the pH of the samples was slightly acidic. Also, others have indicated that pH has a role mainly in the beginning of the of the reaction and has little impact past about 25 min of reaction time.